# High-precision tracking and positioning for monitoring Holstein cattle

**Wei Luo** [1,2,3]*, **Guoqing Zhang** [1], **Quanbo Yuan** [1,4], **Yongxiang Zhao** [1], **Hongce Chen** [1], **Jingjie Zhou** [4,5], **Zhaopeng Meng** [4], **Fulong Wang** [1], **Lin Li** [1], **Jiandong Liu** [1], **Guanwu Wang** [1], **Penggang Wang** [1], **Zhongde Yu** [1]

**1** North China Institute of Aerospace Engineering, Langfang, China, **2** Aerospace Remote Sensing Information Processing and Application Collaborative Innovation Center of Hebei Province, Langfang, China, **3** National Joint Engineering Research Center of Space Remote Sensing Information Application Technology, Langfang, China, **4** College of Intelligence and Computing, Tianjin University, Tianjin, China, **5** Tellyes Scientific Inc. Tianjin, China

* luowei@radi.ac.cn

## Abstract

Enhanced animal welfare has emerged as a pivotal element in contemporary precision animal husbandry, with bovine monitoring constituting a significant facet of precision agriculture. The evolution of intelligent agriculture in recent years has significantly facilitated the integration of drone flight monitoring tools and innovative systems, leveraging deep learning to interpret bovine behavior. Smart drones, outfitted with monitoring systems, have evolved into viable solutions for wildlife protection and monitoring as well as animal husbandry. Nevertheless, challenges arise under actual and multifaceted ranch conditions, where scale alterations, unpredictable movements, and occlusions invariably influence the accurate tracking of unmanned aerial vehicles (UAVs). To address these challenges, this manuscript proposes a tracking algorithm based on deep learning, adhering to the Joint Detection Tracking (JDT) paradigm established by the CenterTrack algorithm. This algorithm is designed to satisfy the requirements of multi-objective tracking in intricate practical scenarios. In comparison with several preeminent tracking algorithms, the proposed Multi-Object Tracking (MOT) algorithm demonstrates superior performance in Multiple Object Tracking Accuracy (MOTA), Multiple Object Tracking Precision (MOTP), and IDF1. Additionally, it exhibits enhanced efficiency in managing Identity Switches (ID), False Positives (FP), and False Negatives (FN). This algorithm proficiently mitigates the inherent challenges of MOT in complex, livestock-dense scenarios.

## 1. Introduction

Livestock husbandry, an integral component of agriculture, involves the raising of livestock and has evolved significantly with the advent of smart agricultural technologies during the fourth agricultural revolution. This intricate practice presents various challenges, including the need for animal traceability, effective health information management, and accurate performance recording. In response to these challenges, innovative monitoring systems

**Data Availability Statement:** All relevant data are within the manuscript and its Supporting information files.

**Funding:** This research was funded by the central government guides local funds for science and

technology development [No. 236Z7201G&No.226Z0302G]; the Special Project of Langfang Key Research and Development under Grant [No. 2023011005B]. Investor (Wei Luo) is responsible for the paper's writing – review & editing.

**Competing interests:** The authors declare that they have no known competing financial interests or personal relationships that could have appeared to influence the work reported in this paper.

incorporating deep learning and drone technology have recently emerged to monitor farm animals and assess their welfare at the farm level. These systems serve multiple purposes, such as evaluating various monitoring systems, diagnosing welfare issues at individual farms, and assisting farm managers in identifying, preventing, or resolving problems related to herd welfare [1]. Recent research surveys have emphasized the importance of allowing cows to express their natural behaviors in their native environment [2–4]. Consequently, the adoption of these systems contributes to improved animal welfare, preservation of pasture, and informed decision making among farm managers, the development of management-related systems, reduced workload, and increased profits.

Unmanned aerial vehicles (UAVs) hold significant promise for wildlife monitoring. Equipped with remote sensors and integrated with positioning technology, UAVs can capture extensive remote sensing data, generating high-definition images with spatial resolutions accurate to centimeters or even millimeters [5]. The UAV remote sensing technology has gradually and extensively applied to investigate the wild animals in grasslands [6–9]. However, drone-based detection faces some challenges. For example, Luo et al. highlighted that when drones are applied to track and monitor dense populations of Proctor antelopes, the presence of individual antelopes obstructing each other can decrease detection accuracy [10]. Moreover, improving detection accuracy often requires increased hardware configuration and computational resources, posing logistical and financial challenges.

Recently, Multi-Object Tracking (MOT) technology has emerged as a prominent area of research in computer vision [11,12]. It involves analyzing video image data collected from visual sensors to extract crucial information about targets, including their appearance features and motion. Additionally, it can explore the relationships among distinct targets simultaneously, ultimately generating continuous motion trajectories for each target. Given its mission-specific characteristics, MOT technology holds remarkable practical significance in distinct fields like autonomous driving [13] and animal monitoring [14]. Currently, deep learning-based MOT technology can be broadly categorized into Tracking by Detection (TBD) and Joint Detection Tracking (JDT), depending on whether the algorithm framework is end-to-end [15].

Most contemporary multi-objective tracking approaches adhere to the tracking by detection paradigm. These methods can be roughly categorized into online methods [11,12,16–23] and offline methods [24–28]. Online methods extend trajectories at each time step, while offline methods update trajectories after processing a batch of frames. Within the TBD framework, the initial step involves employing a target detector [29–32] to locate all objects in each frame. Subsequently, the associations between detections in different frames are established by comparing the similarity of features extracted from various sources, such as motion models [33,34], re-identification models [12,19,20,35], or graph neural networks [18,24,25]. Bewley et al. proposed the SORT algorithm in 2016 [11], which achieved fast speed and high accuracy without occlusion. But it hardly handles occlusion, resulting in high ID switching times and low tracking accuracy in the presence of occlusion. In response to this issue, Wojke et al. optimized the SORT algorithm [11] in 2017 and proposed the Deep SORT algorithm [12], which introduced a deep learning model to extract the appearance features of the target for nearest neighbor matching in real-time target tracking. This improved the target tracking performance in occluded situations and also reduced the problem of target ID switching. The algorithm achieved a significant frame rate in the tracking benchmark. However, the performance of TBD largely depends on the quality of the target detector used. In addition, due to the variety of required feature components, the training process is relatively complex.

In order to reduce computational costs, researchers [36–46] have made significant efforts to use neural networks to complete detection and tracking tasks. For example, Trackor++ [36]

explored the bounding box regression head of object detectors to enhance tracking, proposing the re ID model (Siamese Network) and motion model (motion estimation) to better enhance tracking performance. The chained tracker [38] takes two consecutive frames as inputs and predicts a pair of bounding boxes for the same target. It uses simple IoU information, and two adjacent and overlapping nodes can be linked together through their boxes in the common frame. By alternately applying pairwise box regression and node linking, tracking trajectories can be generated, achieving good tracking results. JDE [42] adopts the Feature Pyramid Network (FPN) architecture [44] to output a dense prediction map, which includes box classification results, box regression coefficients, and dense embedding maps. These methods have been extensively tested and validated for their value in the field of multi-target tracking. But its tracking performance is not ideal when facing occlusion conditions. FairMOT [45] adopted a JDE like approach, but replaced the backbone network with an encoder decoder network, running in a center-based manner, achieving better tracking performance and better handling of occlusion issues. But it also has certain problems, as it combines detection and appearance feature extraction in the network structure, triggering competition among various components and increasing the appearance of target IDs. Based on the FairMOT model, SimpleTrack [47] uses different feature fusion branches to extract different features, aiming to enhance the feature map modeling in the task head network, amplify the differences between the task head and the original feature map, and improve the detection and recognition performance of the algorithm. Wang et al. [48] applied a time-based encoder decoder architecture to multi frame prediction while estimating multi-channel trajectory maps. Although these methods have achieved good detection and recognition performance, they still have not solved the problem of increased ID switching during long-term tracking.

In the research of drone target positioning, determining the three-dimensional coordinates of stationary targets is a relatively straightforward task. This involves utilizing the position and attitude information of the drone in relation to the target, along with angle and range data from the optoelectronic platform, which can be directly fed into the positioning solution model. However, maneuvering targets introduce complexity as they are in constant motion, posing challenges to accurate tracking influenced by various factors. Achieving high-precision target positioning with unmanned aerial vehicles requires addressing the challenge of state estimation for these dynamic targets. The objective is to utilize the observation data to estimate parameters such as the position, velocity, and current state of the target. For nonlinear system estimation, the Particle Filter (PF) algorithm is commonly employed due to its strong filtering performance. However, it may exhibit suboptimal performance when tracking target with high maneuverability. Consequently, researchers have focused on enhancing the PF algorithm with dynamic models. For instance, Magill et al. [49] initially introduced a Multiple Model (MM) algorithm by integrating with multiple filters, each corresponding to different target motion models. Blom et al. [50] analyzed the interaction among multiple models within the MM algorithm in detail and proposed the Interactive Multiple Model (IMM) algorithm by incorporating the Markov transition probabilities to better adapt to changing target behaviors. In 2003, Boers et al. [51] put forward the IMM-based PF algorithm, which demonstrated superior tracking performance in scenarios involving targets with strong maneuverability.

Presently, MOT encounters several challenges. In densely congested scenarios, detection results may not always be completely reliable. Instances of missed detections can lead to inaccuracies in tracking, and the tracking process is susceptible to interruptions when objects are obstructed or temporarily vanish. On the other hand, target association in complex scenes often necessitates the use of re-identification features to extract deep surface features, enabling the re-recognition of partially occluded objects. However, the repetitive nature of feature extraction operations can significantly increase the computational workload, making it

challenging for drone platforms to cope with substantial computational demands. In addition, many visual-based algorithms tend to overlook the height and rolling motion of helicopters. Considering that the camera is typically fixed to the drone, rapid drone movements can cause noticeable changes in field of view of the camera, resulting in reduced tracking performance.

This article presents an effective tracking algorithm that adopts the JDT paradigm building upon the CenterTrack [41] algorithm to address the issues mentioned earlier. The primary contributions of this article can be summarized as follows:

- A feature enhancement module is designed, incorporating the attention mechanism and the Swish Activation Function (SAF). This module boosts the tracking performance, especially of the model for small and overlapping targets.

- A two-stage matching algorithm is proposed to effectively reduce the tracking interruptions and other issues arising from short-term occlusions of objects.

- The IMM-PF algorithm is employed to maintain high-precision target positioning and improve overall target tracking performance.

The structure of this article is organized as follows. Section 2 introduces the research field and objects and provides an overview of the overall system framework. Sections 3 and 4 delve into the presentation and discussion of the experimental results, respectively. Finally, Section 5 summarizes the findings and conclusions.

## 2. Materials and methods

### 2.1 Study area and objects

The study area selected is situated in Youzhi Ranch, Luquan District, Shijiazhuang City, Hebei Province, it is located at 114.35E, 37.98N, covering a total area of 1,206 acres. Currently, the ranch accommodates approximately 5,500 dairy cows, of which 5,000 are purebred Holstein cows. Holstein cows are globally recognized as the top milk-producing and most prolific dairy breed. Originating from the Netherlands, they have been continually refined through breeding programs in various countries due to their exceptional milk production capabilities. Currently, 80%–90% of dairy cows worldwide can trace their lineage back to Holstein cows. This ranch has embraced innovative farming practices, employing the first mobile farming mode of the world and fully automated rotary milking. This pioneering approach has resulted in the production of high-quality milk, reaching international advanced standards in ranch construction, dairy farming, and dairy variety. Consequently, the ranch serves as an ideal setting for investigating precise grazing technology driven by artificial intelligence.

In August 2023, the author and research colleagues conducted aerial photography at Shijiazhuang Youzhi Ranch. A total of 20 aerial sorties were completed at an altitude of 10 meters for sampling. Subsequently, Holstein cows were selected as the research subjects, as depicted in Fig 1. Holstein cows are recognized for their distinctive characteristics, including a tall stature, well-proportioned structure, thin skin, slender bones, and lower subcutaneous fat. They exhibit a more developed hindquarter compared to the forequarters, resulting in a wedge-shaped appearance when viewed from the side. These cows possess distinct characteristics such as short capillaries, black and white patches with well-defined boundaries, white spots on the forehead, and white patches on the lower abdomen, below the knee joints of the limbs, and at the tail.

### 2.2 System overview

To collect data within the selected area, a P600 type intelligent UAV (manufactured by Chengdu Bobei Technology Co., Ltd., China) was adopted. Specific details regarding the UAV

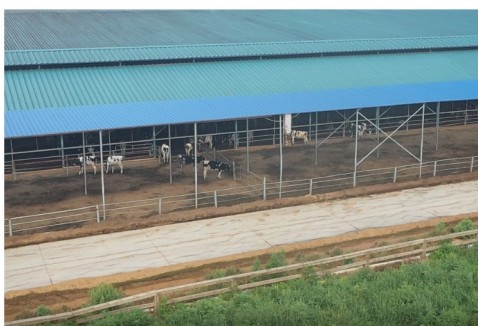 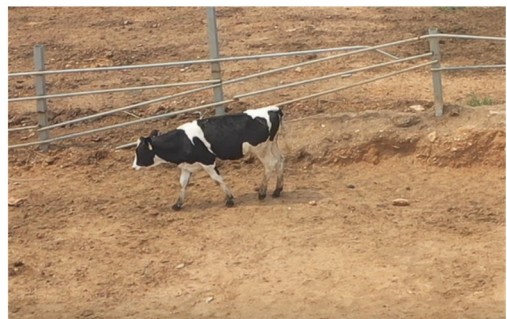

**Fig 1. Aerial images of the study area.**

are presented in S1 Table. The P600 UAV serves as a robust research platform, featuring a substantial payload capacity, extended endurance, and scalability. Equipped with pods, two-dimensional planar Lidar, and GPS, it enables functions such as pod selection and tracking, LiDAR obstacle avoidance, UAV position, and speed-guided flight. In addition, the P600 UAV integrates a Q10F 10x single light pod with a USB interface to enhance its capabilities. A specific robot operating system (ROS2) driver was developed convenient operation with the P600. This equipment allows for the real-time image capture through the airborne computer, achieving image resolutions as fine as 5 cm. Furthermore, it can autonomously track the targets and adjust its position to maintain a constant distance from moving targets. During the target tracking, both the UAV and the pod can operate autonomously via ROS2 control.

The workflow of this system is illustrated in the following Fig 2. Initially, drone camera captured the image of the target. Subsequently, the improved high-reliability tracking algorithm was employed to detect and track the target. In response to the decrease in tracking accuracy resulting from motion between the drone and the target during tracking, a target position estimation model was designed herein. This model applies the IMM-PF algorithm to maintain high-precision target positioning, thereby enhancing the overall target tracking performance.

## 2.3 Tracking algorithm

The proposed tracking algorithm primarily comprises four key components: input, backbone network, output, and matching (two-stage). The input data includes red-green-blue (RGB)

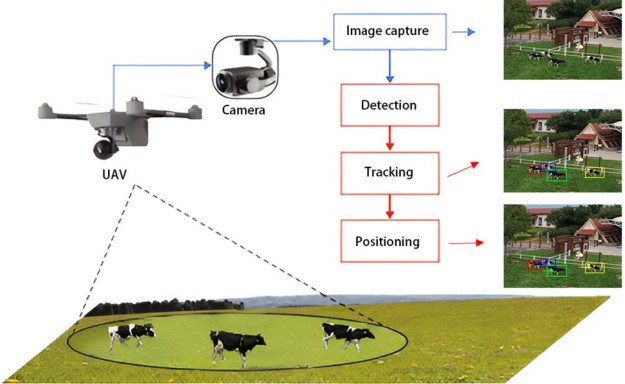

**Fig 2. The overall technical framework proposed.**

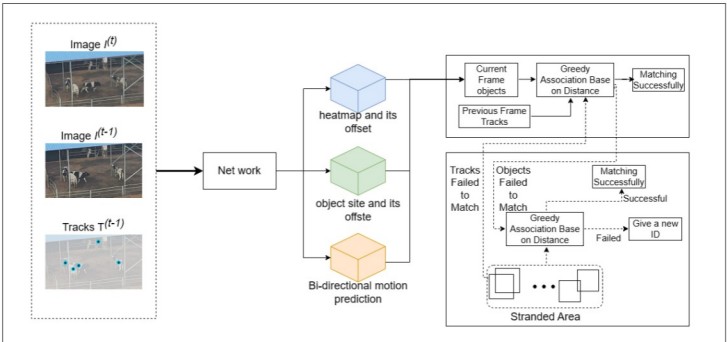

**Fig 3. MOT pipeline.**

images from both the current and previous frames, along with the heat-map generated by the model based on the previous frame. Furthermore, the modified DLA-34 served as the backbone network for feature extraction. Subsequently, the obtained feature maps were input into three branches: heat-map and its offset, object size and its offset, and bi-directional motion prediction. The outputs from these branches play crucial roles in two matching stages. The overall workflow is visually depicted in Fig 3.

**2.3.1 Improved backbone network.** The DLA-34 network presents an encoder decoder structure that utilizes multi-layer feature aggregation to fuse shallow and deep features. However, the original DLA-34 had a complex network structure and many parameters, consuming a large amount of computing resources. In addition, the original DLA-34 network has poor detection and tracking performance for small and occluded targets due to insufficient feature extraction capabilities. In order to reduce the number of network computing parameters, improve computational efficiency, and enhance network generalization ability, we have made two key improvements [52] to the backbone network of the algorithm based on the DLA-34 network.

Firstly, the Pyramid Segmentation Attention (PSA) feature enhancement module and the SAF were integrated into the residual block of the DLA-34 network. Secondly, the network structure and the number of layers were adjusted as required. As explicated in Fig 4, the optimized backbone network is composed of 6 layers (layers 0–5), facilitating the entire downsampling process. Initial feature extraction from the input image is completed in layers 0 and 1 using traditional two-dimensional convolution normalization and activation functions. Layers 2–5 utilize an iterative deep aggregation structure to comprehensively extract the feature data

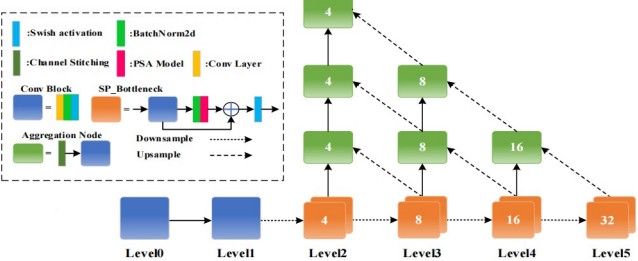

**Fig 4. Backbone network diagram.**

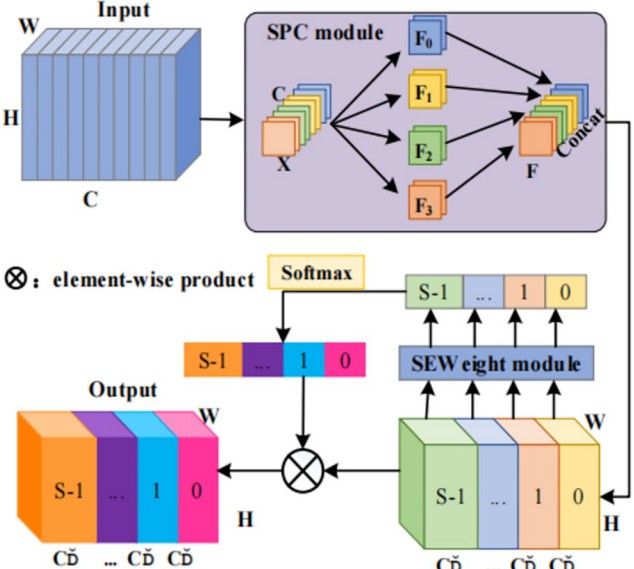

**Fig 5. The structure of PSA module.**

of the target. Layers 3–5 in Fig 4 represent the up-sampling process. This article interleaves and connects various basic modules and aggregation nodes to form a tree-like unit with the ultimate goal of facilitating information exchange among varying levels. For this purpose, the PSA model [53] was integrated into the backbone network by replacing the 3*3 convolutional kernel in the node residual block, as depicted in Fig 4. In addition, this article replaced the Relu activation function adopted in the original DLA-34 network with the SAF to enhance the generalization ability of the model.

As depicted in Fig 5, the PSA module was primarily executed five steps. Initially, the SPC module was employed for channel segmentation, and multi-scale features were extracted based on the spatial information within the feature map of each channel. Subsequently, the SEW eight module was utilized to extract channel attention from feature maps at distinct scales, yielding channel attention weights corresponding to these scales. Then, the Softmax activation function was applied to recalibrate the multi-scale channel attention vector, resulting in new attention weights after interaction across these different scales. Moreover, the recalibrated weights were element-wise multiplied by their respective feature maps, generating a feature map enriched with multi-scale feature information and attention weights. Lastly, the refined feature map containing a wealth of multi-scale feature information is output by the model.

Substituting the 3*3 convolution kernel within the residual block of the node using the PSA module can facilitate the incorporation of multi-scale spatial information and cross-channel attention into every individual node of the DLA-34 backbone network.

In the mode for the original DLA-34 backbone network, the residual block is referred to as the Basic Block and its addition strategy is illustrated in Fig 5. The Basic Block, along with its addition approach, is replaced by the PSA module, which substitutes the 3*3 convolution kernel of the Bottleneck in the original ResNet [54]. The experimental results undeniably demonstrate that integrating the Basic Block with the PSA module enhances the tracking performance significantly within the DLA-34 architecture.

SAF is characterized by its smooth, continuous, and monotonic attributes, with a lower but no upper bound. Incorporating SAF into a model results in a more gradual gradient propagation, preservation of valuable information, and enhancement of the model's generalization ability. The incorporation of SAF into our model proves advantageous for enhancing its target detection capacity, especially in scenarios involving overlapping targets. The mathematical expressions for the SAF and its derivatives are given in Eqs (1) and (2).

$$Swish = x \cdot \sigma(\beta x) \tag{1}$$

$$Swish' = \beta \cdot x \cdot \sigma(\beta x) + \sigma(\beta \cdot x)(1 - \beta \cdot x \cdot \sigma(\beta x)) \tag{2}$$

Where $\sigma$ is the sigmoid function, and $\beta$ signifies a constant or trainable parameter.

**2.3.2 Two-stage matching algorithm.**  The baseline algorithm CenterTrack uses a distance based greedy matching algorithm to complete data association. It can track targets in some scenarios, but in some complex and dense scenarios, this method cannot effectively track occluded targets for a long time, resulting in a sharp increase in ID switching times and a decline in tracking performance.

A two-stage matching algorithm is designated [55] in this article to improve the tracking performance in the case of object occlusion. The algorithm leverages bi-directional motion information and a given stranding region to execute two distance-based greedy matching processes. Tracking boxes that do not find a match in the first matching phase are relocated to the stranding area, where their positions are constantly updated based on object motion prediction. Therefore, when these objects appear again, they can be successfully matched in the second matching phase.

The proposed matching algorithm consists of two stages, both employing distance-based greedy matching strategies. In the first matching phase, objects with successful matches inherit their ID information from the previous frame, while unmatched objects are kept in the standing area for potential second-stage matching.

Failure of the initial match can be attributed to two conditions: when a new object emerges in the current frame, and when an object that was blocked in the previous frame appears again. To mitigate the risk of incorrectly identifying re-emerging objects as new entities, a significant challenge in MOT, this article introduces a "stranding area" to temporarily accommodate the obscured objects, providing them an opportunity for a second match if they reappear. To enhance the implementation of the distance-based greedy algorithm, targets in the stranded frame were constantly displaced using motion vectors relative to future frames, simulating the movement of the target after occlusion in the real world. Therefore, when these targets reappear, they can be reconnected through a distance-based greedy matching algorithm, ensuring the continuity of the tracking trajectory. The visualization of this process is depicted in Fig 6.

The duration for which an object remains within the "stranding area" is contingent upon the timing of its reappearance. In some cases, some objects may not reappear at all, necessitating their removal from the stranding area. To identify objects that are unlikely to return, each object was allocated a health point upon entering the "stranding area", with the score gradually decreasing over time. When the lifetime value of these objects reached 0, they were subsequently deleted from the "stranding area".

If an unmatched object from the first stage still failed to be matched in the stranding area in the second stage, it was allocated a new ID and treated as a distinct object. The comprehensive matching algorithm is illustrated in S2 Table.

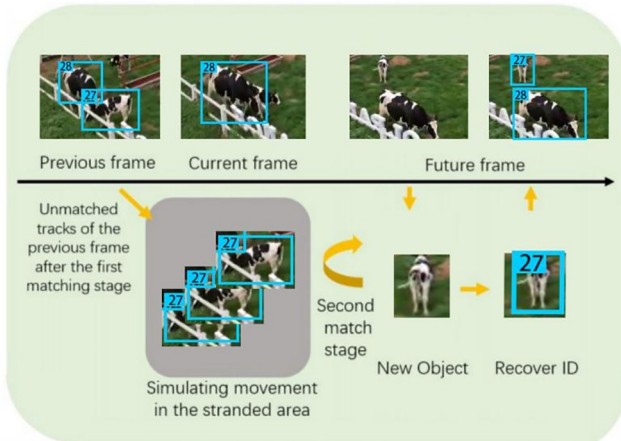

**Fig 6. Operation process in stranding area.**

## 2.4 Target positioning algorithm

During the detection and tracking of ground target s by the UAV, the target is surrounded by a detection frame and marked with an ID. The center of the detection frame is taken as the target point, and its pixel coordinate is $(x_p, y_p)$. The target line-of-sight vector, $\rightarrow r$, is defined as the vector between the optical center of the camera and the target point. As a result, $\rightarrow r$ can effectively reflect the relative position between the target point T and the UAV. The relationship between the parameters is shown in Fig 7.

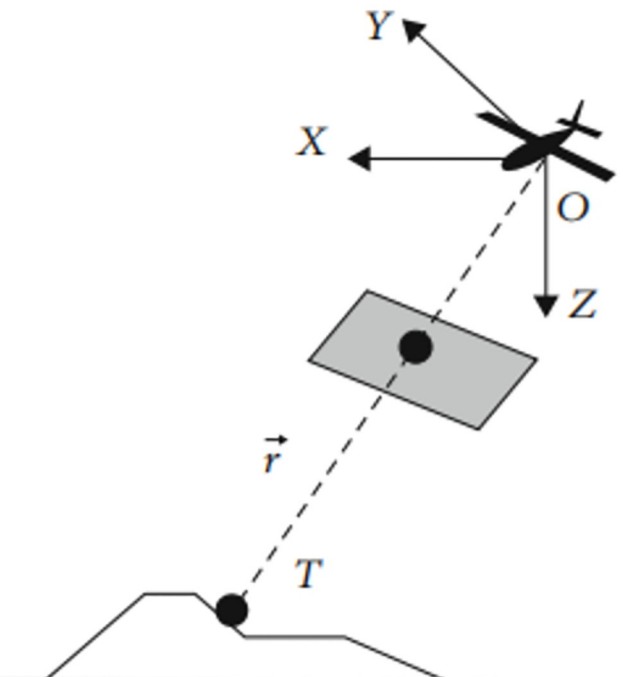

**Fig 7. Schematic showing the line-of-sight angle.**

Geographic coordinate system for the UAV was established with the center of the GPS receiver serving as its origin point herein. In this system, the X-axis and Y-axis point directly north and east, respectively, while the Z-axis complements the system as the vertical axis, forming a right-hand coordinate system. The line-of-sight angle is defined as $(\rho, \varepsilon)$, where $\rho$ signifies the angle between the line-of-sight vector directed towards the target and the Z-axis and is often referred to as the line-of-sight height angle; $\varepsilon$ denotes the angle between the projection of the line-of-sight vector onto the XOY plane and the X-axis, known as the field of view direction angle. Throughout the flight, the attitude angle of UAV, camera pointing, and target position can be integrated to determine the $\rho$ and $\varepsilon$ values.

To compute the target line-of-sight angle, three coordinate systems were assigned: the camera system (referred to as C, with its origin at the camera's optical center), the Inertial Measurement Unit (IMU) system (denoted as I, taking its origin at the IMU measurement center), and the UAV geographic system (knowns as L, with its origin at the GPS receiver center). The spatial interrelations among these coordinate systems are visually illustrated in Fig 8.

In practical situations, the state of cows undergoes dynamic changes, which are difficult to capture using a single motion model. The original baseline algorithm is better at handling linear motion, but in the work of tracking cows using drones, the motion between drones and cows is usually non-linear, which leads to poor tracking performance of the algorithm. In addition, during the process of tracking and detecting cows, the horizontal, vertical, and rotational movements of the drone camera can also affect the tracking effect of the cows. Therefore, it is necessary to establish a drone positioning algorithm to reduce the interference caused by the rapid movement of drones and targets on tracking, achieve precise target positioning, and assist drones in better tracking targets. The task of drone target positioning encompasses three main systems: the aircraft itself, the camera, and the GPS/Inertial Navigation System (INS).

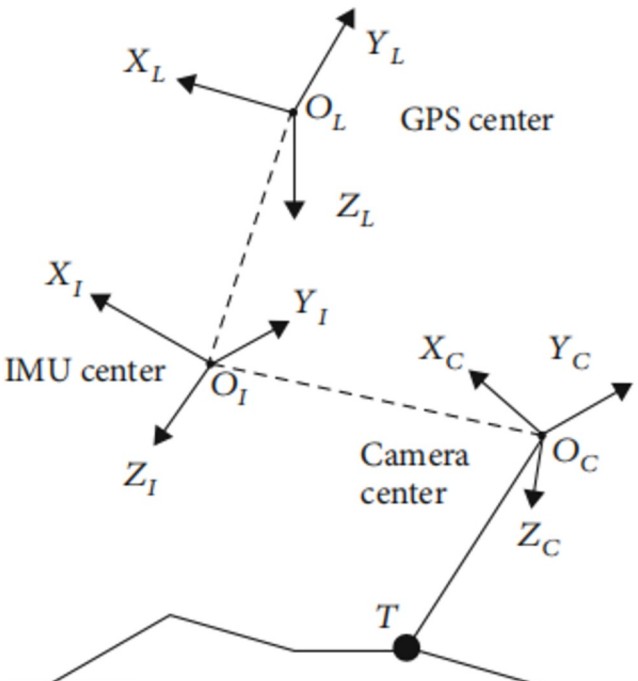

**Fig 8. Coordinate conversion.**

However, these systems come with their own inherent errors. GPS measurements may exhibit inaccuracies in estimating the aircraft's latitude and longitude, while INS can introduce errors in measuring the aircraft's attitude. Additionally, camera visual axis jitter further compounds the complexities. When dealing with moving targets, the PF algorithm offers a suitable approach, as it excels in handling nonlinear and non-Gaussian system filtering. In this article, the combined strengths of the IMM and PF algorithms were harnessed into the IMM-PF algorithm to achieve precise target localization [51].

In the context of multiple models, the equations for state transition and observation are expressed as follows:

$$x_k = F(m_k)x_{k-1} + G(m_k)u_{k-1}(m_k), \tag{3}$$

$$z_k = H(x_k, m_k) + v_k(m_k). \tag{4}$$

In the above equations, $x_k$ represents the target state vector of the model $m_k$ at time $k$, and $z_k$ signifies the corresponding state observation variable. The state transition matrix ($F$), the observation matrix ($H$), the process noise ($u_k$), and the observation noise ($v_k$) are all associated with the model $m_k$. The probability densities of $u_k$ and $v_k$ are defined as $d_{u_{k(m_k)}}(u)$ and $d_{v_{k(m_k)}}(v)$, respectively.

The IMM-PF algorithm tool the IMM algorithm as its foundational framework, incorporating the PF as the model matching filter. The IMM algorithm encompass four distinct steps: input interaction, model matching filtering, model probability update, and estimated output. Within the framework of the IMM algorithm, recursive Bayesian filtering was employed to describe evolution of the IMM-PF algorithm from time k-1 to k.

(1) Inputting interaction: the interaction probability of the model at time k-1 was computed by utilizing the expression (5):

$$\mu_{k-1}(m_{k-1} \mid m_k) = \frac{p_{i,j}\mu_{k-1}(m_{k-1})}{b_{k-1}(m_k)}. \tag{5}$$

The normalization factor was expressed as follows:

$$b_{k-1}(m_k) = \sum_{m_{k-1} \in M} p_{i,j}\mu_{k-1}(m_{k-1}). \tag{6}$$

Interactions for state estimations among different particles in varying models $l = (1, 2, \cdots, N)$ were as follows:

$$\tilde{x}_{k-1}^l(m_k) = \sum_{m_{k-1} \neq m_k}^{M} \tilde{x}_{k-1}(m_{k-1})\mu_{k-1}(m_{k-1} \mid m_k) + \tilde{x}_{k-1}^l(m_k)\mu_{k-1}(m_k \mid m_k). \tag{7}$$

(2) Interactive model matching filtering: the particle state at time k was predicted by Eq (3):

$$\tilde{x}_k^l(m_k) = F(m_k)\tilde{x}_{k-1}^l(m_k) + G(m_k)\tilde{\mu}_{k-1}^l(m_k) \tag{8}$$

The observed value of the particle state at time k was predicted by Eq (4):

$$\tilde{z}_k^l(m_k) = H(\tilde{x}_k^l, m_k) \tag{9}$$

The particle weight was determined based on the observed system state, denoted as $z_k$, and the probability density of observation noise was designated as $d_{v_{k(m_k)}}(v)$.

$$\tilde{w}_k^l(m_k) = d_{v_{k(m_k)}}(z_k - \tilde{z}_k^l(m_k)) \tag{10}$$

The normalized weight was expressed as follows:

$$\tilde{w}_k^l(m_k) = \frac{\tilde{w}_k^l(m_k)}{\sum_{l=1}^{N} \tilde{w}_k^l(m_k)} \tag{11}$$

In the above equation, $\tilde{x}_k^l(m_k)$ was subjected to a further sampling using the $[\bar{x}_k^l(m_k) = \tilde{x}_k^l(m_k)] = \tilde{w}_k^l(m_k)$, yielding a new particle set $\bar{x}_k^l(m_k)$, with the particle weight of $\bar{x}_k^l(m_k) = \frac{1}{N}$. In this context, state of the model $m_k$ at time k can be estimated as follows:

$$\tilde{x}_k^l(m_k) = \frac{1}{N} \sum_{l=1}^{N} \bar{x}_k^l(m_k) \tag{12}$$

(3) Update of the model probability: the residual of particle observation was expressed in Eq (13) below:

$$r_k^l(m_k) = z_k - H(\bar{x}_k^l, m_k) \tag{13}$$

Similarly, mean of the particle observations could be calculated by expression below:

$$\bar{z}_k(m_k) = \frac{1}{N} \sum_{l=1}^{N} H(\bar{x}_k^l, m_k) \tag{14}$$

At this time, the residual covariance was acquired as below:

$$S_k(m_k) = \frac{1}{N} \sum_{l=1}^{N} [H(\bar{x}_k^l, m_k) - \bar{z}_k(m_k)] \cdot [H(\bar{x}_k^l, m_k) - \bar{z}_k(m_k)]^T \tag{15}$$

The likelihood function was represented in below equation:

$$\Lambda_k(m_k) = \frac{1}{N} \sum_{l=1}^{N} N(r_k^l(m_k); 0, S_k(m_k)) \tag{16}$$

Ultimately, the model probability was updated:

$$\mu_k(m_k) = \frac{\Lambda_k(m_k) b_{k-1}(m_k)}{B_k} \tag{17}$$

$$B_k = \sum_{m_k \in M} \Lambda_k(m_k) b_{k-1}(m_k) \tag{18}$$

(4) Estimated output: state of the target was assumed as below expression:

$$\hat{x}_k = \sum_{m_k \in M} \hat{x}_k(m_k) \mu_k(m_k) \tag{19}$$

## 3. Results

### 3.1 Metrics for tracking

This article assesses various MOT algorithms in diverse scenarios, employing the following evaluation criteria.

MOT accuracy (MOTA) serves as an intuitive gauge for evaluating the performance of object detection and trajectory maintenance, regardless of the estimated accuracy of object positions. A higher MOTA value signifies superior performance, which could be computed with the following equation:

$$MOTA = 1 - \frac{\sum FN_t + FP_t + IDSW_t}{\sum_t GT_t} \tag{20}$$

where, $FN_t$ and $FP_t$ suggested false negative and false positive, respectively; $IDSW_t$ is ID Switch, and $GT_t$ denotes the number of all objects.

MOT precision (MOTP) was adopted to judge the positioning precision, with a larger MOTP values suggesting a better positioning effect.

$$MOTP = \frac{\sum_{t,i} d_{t,i}}{\sum_t C_t} \tag{21}$$

where, d was the average metric distance (i.e., the IoU value of the bounding box) and C denoted the number of current frames which were successful matched.

IDF1 referred to the F1 score for the identification, which is calculated using Eq (22), where IDTP indicates the number of correctly matched identifications, while IDFP and IDFN denote the number of incorrectly matched and unmatched identifications, respectively.

$$IDF1 = \frac{2 \times IDTP}{2 \times IDTP + IDFP + IDFN} \tag{22}$$

Mostly tracked (MT) refers to the count of successful tracking results where the object's position matches the ground truth for at least 80% of the time.

Mostly lost (ML) signifies the count of successful tracking results where the object's position matches the ground truth for less than 20% of the time.

ID switch denotes the count of instance in which assigned IDs change during tracking.

Fragmentation (FM) indicates the number of times tracking is interrupted, meaning instances when the tagged object fails to be matched.

FP represents the count of false alarms, which corresponds to the incorrect trajectory predictions.

FN reflects the count of missed detections and undetected tracking objects.

### 3.2 Dataset and experiment setups

In our experiment, the intelligent unmanned field platform equipped with the Jetson AGX Xavier developed by NVIDIA was utilized for onboard image processing. This modular super-computer boasts a formidable configuration, featuring a 512 CUDA-core NVIDIA Volta GPU along with an 8-core ARMv8.2 CPU, offering remarkable AI computing capabilities. Notably, it exhibits a 10-fold increase in power consumption ratio and a 20-fold boost in performance when compared to the previous Jetson TX2 platform, which was equipped with a 256 CUDA-core NVIDIA Pascal GUP and a CPU of quad-core ARM.

In this article, a dataset was curated, consisting of the data of 10 cow video sequences from the study area. Approximately 100 video frames with the same interval were selected as a batch, which were converted into JPEG images of uniform size (640×640), obtaining 6,000 cow images in total. Subsequently, these images were annotated using Labelimage to save them as XML files. Next, they were annotated into three subsets: training, validation, and testing at a ratio of 7:2:1. The model was trained based on CenterNet, using the improved DLA-34 as the backbone network. Meanwhile, the model optimization was conducted using the Adam optimizer, with the following training parameters: an epoch count of 160, a batch size of 24, and an initial learning rate of $1\times10^{-4}$, which was subsequently reduced to 1/10 during the $100^{th}$ and $140^{th}$ epochs. In addition, considering that more than 70% of the occlusion duration in the cow dataset curated is less than 20 frames, the initial lifespan value of the objects in the "stranding area" is set to 20.

## 3.3 Evaluation of benchmarks

To validate the effectiveness of the algorithm enhancements proposed in this article and to explore the most impactful improvements, a series of ablation experiments were conducted on the optimized CenterTrack-based MOT algorithm.

Firstly, ablation experiments were performed. MOTA, IDF1, and IDs were selected for comparative testing to demonstrate the individual contributions of varying improvements, as summarized in Table 1. (Note: ↑ signifies that a higher evaluation index value indicates better performance; while ↓ suggests that a lower value means better performance).

Table 1 reveals that compared to the original network, the introduction of the PSA module resulted in an increase of 6.1% and 0.5% in MOTA and IDF1, respectively, while reducing the IDs by nearly 10%. By introducing the SAF, the MOTA and IDF1 were increased by 1.3% and 1.6%, respectively, and the IDs also decreased to some extent. In addition, IDs were significantly reduced through employing the two-stage matching algorithms in place of the distance-based greedy matching algorithms. This experiment demonstrated that in contrast to the baseline method, the improved approach substantially optimized the MOTA, IDF1, and IDs, indicating its enhancements in accuracy of target discrimination and tracking while delivering a notable optimization effect on the tracking algorithm.

Furthermore, the tracking results under two different scenarios were visualized and analyzed in Figs 9 and 10, respectively, to assess the robustness of our algorithm in various scenarios.

In Fig 9, it was apparent that the cow within the red elliptical box was not detected using the CenterTrack tracking algorithm in frame 450, but it was assigned an ID of 215 by frame 455. By frame 460, both the detection box and ID of the cow disappeared once more due to occlusion. Conversely, in the same scenario, our algorithm effectively detected cows and stably assigned them an ID of 198. In addition, the CenterTrack failed to detect all small targets in the image. Consequently, our algorithm benefits from the finer-grained features provided by

**Table 1. Ablation experiment of baseline algorithm.**

| Model | MOTA ↑ (%) | IDF1 ↑ (%) | IDsw ↓ |
|---|---|---|---|
| Baseline | 62.4 | 66.7 | 285 |
| Baseline (DLA-34+PAS) | 68.5 | 67.2 | 259 |
| Baseline (DLA-34+PAS+Swith) | 69.8 | 68.8 | 256 |
| Baseline+ two-stage matching algorithm | 70.5 | 69.4 | 243 |
| Ours | 72.7 | 70.2 | 238 |

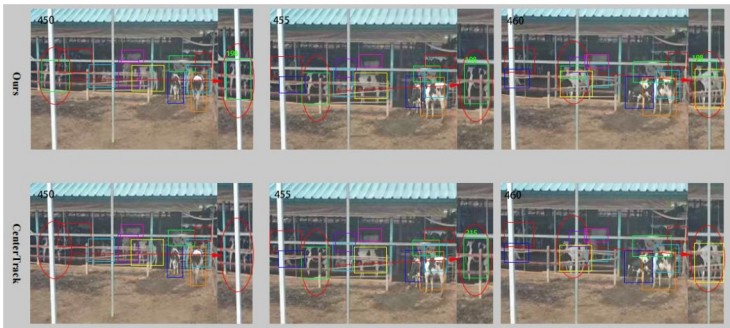

**Fig 9. Comparison of tracking results between our tracking algorithm and the CenterTrack in complex scenarios.**

the PSA module, demonstrating strong detection and tracking performance for cows in the image.

Fig 10 provides a detailed view of a specific object tracking when the object is occluded and then reappears. In this case, target 12 was almost completely obscured by target 16 in the intermediate frame. As depicted in Fig 10(a), when target 12 appeared again, the CenterTrack algorithm assigned it a new ID, namely, 20. However, our algorithm temporarily retained the occluded target 15 in the stranding area and restored its ID when it reappeared (Fig 10(b)), thus ensuring uninterrupted tracking.

Finally, the target localization algorithm was integrated to our tracking algorithm for experiments to compare it with the original tracking algorithm, thus assessing the effectiveness of the target localization algorithm in improving the performance of our tracking algorithm. With MOTP as the evaluation indicator, the evaluation results are outlined in Table 2.

Table 2 demonstrates a notable enhancement in the MOTP value for the tracking algorithm integrated with the target localization algorithm, further validating the satisfactory performance of the target positioning algorithm.

Fig 11 illustrates the tracking results of a video sequence, showcasing the impressive performance of our tracking algorithm in challenging environments. Therefore, this algorithm excels in accurately detecting multiple targets in each frame, while maintaining consistent tracking of the same targets.

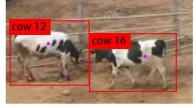 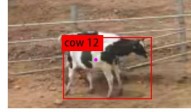 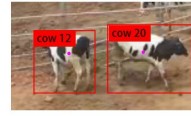

(a) CenterTrack algorithm

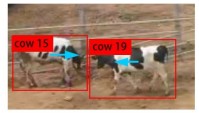 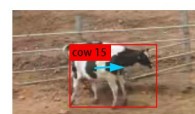 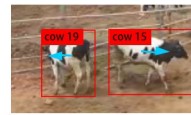

(b) Our tracking algorithm

**Fig 10. An example of tracking occluded objects using (a) CenterTrack algorithm; (b) Our algorithm; The blue arrow represented the bidirectional motion vector.**

**Table 2. Performance comparison between the original tracking algorithm and the tracking algorithm with target localization algorithm.**

| Model | MOTP ↑ (%) |
|---|---|
| Ours (Without target localization algorithm) | 62.4 |
| Ours (With target localization algorithm) | 67.5 |

Table 3 provides a comparison of the tracking performance between the proposed tracking algorithm and the most advanced tracking algorithm.

Table 4 provides a comparison of tracking performance between the proposed tracking algorithms and state-of-the-art tracking algorithms on the MOT17 and MOT20 datasets.

## 4. Discussion

The experimental observations demonstrate several advantages of our tracking algorithm. (a) It incorporated attention mechanisms and multi-scale fusion using a pyramid segmented attention model to improve the network performance in detecting occluded targets. Furthermore, the ReLU activation function was replaced with SAF to enhance the generalization ability and to effectively improve the tracking accuracy of the algorithm adopted in this article. (b) A two-stage matching algorithm, instead of the distance-based greedy matching algorithm, was employed in the baseline model, which significantly improved the performance in detecting target occlusion. In many cases where a substantial portion of a target is occluded or temporarily disappears from the image, tracking interruptions are common for most existing tracking algorithms. In contrast, the two-stage matching algorithm proposed in this article leverages the bi-directional motion prediction information, improving occlusion processing capabilities. In the matching algorithm, a "stranding area" was set to temporarily store objects that failed be tracked. When these objects appeared, our method firstly attempted to match them with objects in the "stranding area", preventing the mistake of identifying new identities and thus ensuring a more continuous trajectory. (c) Our tracking algorithm was optimized by integrating a target positioning algorithm (IMM-PF algorithm) that can handle nonlinear and non-Gaussian system filtering. In this way, it alleviated errors in estimating the latitude and longitude of aircraft in GPS measurements, errors in measuring aircraft attitude in INS, and

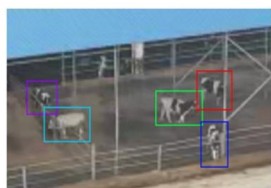
(a) Frame 180 (b) Frame 190

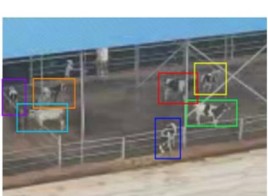 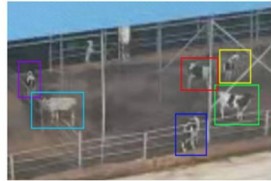
(c) Frame 205 (d) Frame 220

**Fig 11. The tracking results using the target localization algorithm.**

**Table 3. Comparison of our method with other popular models.**

| Method | MOTA↑(%) | MOTP↑(%) | IDF1↑(%) | MT↑(%) | ML↓(%) | FP↓ | FN↓ | IDsw↓ | FPS↑ |
|---|---|---|---|---|---|---|---|---|---|
| SORT | 57.4 | 70.2 | 46.7 | 21.1 | 35.3 | 4303 | 17262 | 402 | 56.0 |
| Deep SORT | 59.3 | 73.2 | 58.2 | 22.8 | 36.3 | 4486 | 15529 | 345 | 7.5 |
| JDE | 63.4 | 74.0 | 62.8 | 28.4 | 34.2 | 5863 | 14568 | 352 | 17.3 |
| FairMOT | 65.9 | 76.4 | 64.3 | 31.2 | 33.6 | 6465 | 13566 | 343 | 21.6 |
| CenterTrack | 67.2 | 75.1 | 64.7 | 33.5 | 31.8 | 3856 | 14981 | 285 | 16.3 |
| Ours | 72.7 | 79.4 | 70.2 | 37.4 | 29.7 | 4635 | 11594 | 238 | 18.7 |

**Table 4. Comparison of our method with other popular models on MOT17 and MOT20.**

| Dateset | Tracker | MOTA↑(%) | IDF1↑(%) | HOTA↑ | IDsw↓ | FPS↑ |
|---|---|---|---|---|---|---|
| **MOT17** | SORT | 42.4 | 38.9 | 33.7 | 4796 | 112 |
| | Deep SORT | 45.53 | 42.8 | 40.2 | 4315 | 14 |
| | JDE | 63.0 | 59.5 | 45.5 | 4172 | 18.8 |
| | FairMOT | 73.7 | 72.3 | 59.3 | 3303 | 20 |
| | CenterTrack | 67.8 | 64.7 | 52.2 | 3039 | 17 |
| | ours | 74.7 | 73.1 | 60.6 | 3002 | 22 |
| **MOT20** | FairMOT | 61.8 | 67.3 | 54.6 | 5243 | 8.9 |
| | ours | 66.6 | 68.8 | 58.3 | 3961 | 9.3 |

issues related to drone camera line-of-sight jitter. As a result, it improved the accuracy in multi target localization.

In addition to the baseline algorithm (i.e., CenterTrack), there are several advanced tracking methods that share similarities with our algorithm. However, they all are subjected to certain drawbacks. For example, SORT achieves the fastest track but lacks appearance features, resulting in lower tracking accuracy. DeepSORT introduces re-recognition features in the correlation section of the data to extract deep surface features, enabling the re-recognition of partially occluded objects. However, it requires repeated feature extraction operations, increasing the computational complexity. JDE simultaneously extracts the Re-ID information (low dimensional vector information) from both the detection frame and the objects within it. Nevertheless, anchors generated based on the Anchor-based detectors may not be suitable for learning appropriate Re-ID information, resulting in a single object identified by several anchors, leading to severe network ambiguity. In addition, the FairMOT algorithm combines detection and appearance feature extraction within a network structure, triggering competition among various components, thereby increasing the occurrence of target identity switches during the tracking.

Table 1 shows that compared to the original network, the introduction of the PSA module resulted in an increase of 6.1 and 0.5 percentage points in MOTA and IDF1, respectively, while ID decreased by nearly 10 percentage points. These performance improvements demonstrate the effectiveness of integrating the PSA module into the backbone network, enhancing the model's ability to detect and track small and occluded targets. By introducing SAF, MOTA and IDF1 have increased by 1.3 and 1.6 percentage points respectively, while ID has also decreased. These performance improvements demonstrate the effectiveness of replacing the Relu activation function with SAF, which enhances the model's generalization ability and enables it to better adapt to different datasets. In addition, replacing distance based greedy matching algorithm with two-stage matching algorithm significantly reduces ID, which proves

that our proposed two-stage matching algorithm performs better in dealing with long-term occlusion target tracking problems. It can reduce ID switching and achieve continuous tracking. In summary, this experiment shows that compared with the baseline method, the improved method significantly optimizes MOTA, IDF1, and ID, indicating that it improves the accuracy of target recognition and tracking, and has a significant optimization effect on the tracking algorithm.

Table 2 shows that compared with the original tracking algorithm, the MOTP value of the tracking algorithm integrated with the target localization algorithm is significantly improved. This performance improvement indicates that our target localization algorithm has better localization performance, thus proving that the IMM-PM algorithm can serve as a better auxiliary tracking for the target localization algorithm and verifying its satisfactory performance.

Table 3 summarizes the comparison results of our tracking algorithm with other mainstream algorithms on our cow dataset. Our tracking algorithm achieved 72.7%, 79.4%, 70.2%, 37.4%, 29.7%, 4635%, 11594%, 238%, and 18.7% of MOTA, MOTP, IDF1, MT, ML, FP, FN, IDsw, and FPS on our cow dataset, respectively. Compared with the baseline algorithm CenterTrack, the MOT algorithm used in this article increases the MOTA value by 5.5%, the MOTP value by 4.3%, the IDF1 value by 5.5%, the FN value by 3387, and the ID is reduced to obtain a higher FPS value. In addition, compared with SORT, DeepSORT, JDE, and FairMOT, the proposed MOT algorithm achieves higher values in MOTA, MOTP, and IDF1, and lower values in IDsw and FN. Due to the fact that SORT algorithm only uses IOU matching, its processing speed is much faster than our algorithm. In addition, due to the integration of PSA and other modules into the algorithm backbone network, our algorithm has achieved detection and recognition of small and occluded targets, reducing the number of missed detections (FN). However, this improvement has also led to an increase in the number of false positives (FP) in our algorithm. In summary, the experimental results demonstrate that our algorithm has achieved the best overall performance, which further confirms the progress and effectiveness of this paper.

Table 4 summarizes the comparison results of our algorithm with other mainstream algorithms on the public datasets MOT17 and MOT20. Compared with mainstream algorithms such as SORT, DeepSORT, JDE, FairMOT, CenterTrack, etc., our algorithm has the best performance among these algorithms except for FPS performance which is not as good as SORT algorithm (SORT algorithm only matches through IOU, so its speed is very fast). This proves that our algorithm can be applied to different datasets, further demonstrating the robustness of our algorithm.

## 5. Conclusion

This article proposes a highly reliable drone target tracking system to address the challenges associated with missed detection and tracking failures in dense scenes where objects often overlap. In terms of algorithm model, this article uses CenterTrack algorithm as the baseline algorithm for the tracker. On this basis, we combined the feature enhancement module and introduced SAF to improve the detection and tracking performance of small and overlapping targets. We propose a two-stage matching algorithm that combines distance based greedy matching with stranded regions to alleviate tracking interruptions caused by short-term occlusion. In addition, we will apply the IMM-PF algorithm to target localization of unmanned aerial vehicles to improve the accuracy of target localization. In terms of algorithm performance, we conducted tracking experiments on Holstein cows in the pasture. The experimental results show that our algorithm achieves 72.7%, 79.4%, 70.2%, 37.4%, 29.7%, 4635, 11594, 238, and 18.7% for MOTA, MOTP, IDF1, MT, ML, FP, FN, IDsw, and FPS, respectively. Compared

with the baseline algorithm CenterTrack, our algorithm has achieved better performance in all aspects except for FP, which proves that our algorithm can effectively track and monitor cattle herds in complex and dense scenes. In addition, we also tested the performance of the algorithm on the public datasets MOT17 and MOT20. The experimental results show that, apart from FPS, our algorithm performs better than mainstream algorithms such as SORT, further proving the robustness of our algorithm.

On the other hand, the methods used in this article have the following limitations that need to be noted. (a) The loss function has not been optimized (b), and the tracking effect is not ideal in environments with changing lighting conditions; (c) The proposed framework aims to track multiple cows in the pasture. Both cows and drone cameras are in motion, but the MOT algorithm does not consider camera motion compensation, which to some extent hinders accuracy.

In future work, we will use a center loss function applied in the field of facial recognition, which not only includes the distance between classes, but also considers reducing intra class differences, achieving inter class separability and intra class compactness, to better solve the problem of high similarity and difficulty in distinguishing between individual cows. In addition, we will integrate visible and infrared images and use the RGB-T algorithm with fusion attention mechanism for tracking to solve the challenge of a single sensor being unable to obtain accurate information due to lighting issues in complex environments such as low light and strong light. This will reduce the possibility of tracking faults and improve robustness under various lighting conditions. At the same time, we will explore integrating global motion compensation into the MOT algorithm, which we consider as an image registration method suitable for revealing background motion. This method extracts image key points and sparse optical flow for feature tracking based on local outlier suppression by translation, thereby compensating for the motion of the drone camera and improving the accuracy of the MOT algorithm.

## Supporting information

**S1 Table. Hardware parameters.**
(PDF)

**S2 Table. Algorithm.**
(PDF)

**S1 Dataset.**
(7Z)

## Author Contributions

**Conceptualization:** Guoqing Zhang.

**Data curation:** Lin Li.

**Formal analysis:** Jiandong Liu.

**Funding acquisition:** Zhaopeng Meng.

**Investigation:** Hongce Chen.

**Methodology:** Yongxiang Zhao, Fulong Wang.

**Project administration:** Penggang Wang.

**Resources:** Guanwu Wang.

**Software:** Zhongde Yu.

**Validation:** Jingjie Zhou.

**Writing – original draft:** Quanbo Yuan.

**Writing – review & editing:** Wei Luo.

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
