## [Decision Letter · Decision Letter 0]

1 Feb 2024

PONE-D-23-43870High-precision tracking and positioning drone for monitoring Holstein cattlesPLOS ONE

Dear Dr. Luo,

Thank you for submitting your manuscript to PLOS ONE. After careful consideration, we feel that it has merit but does not fully meet PLOS ONE’s publication criteria as it currently stands. Therefore, we invite you to submit a revised version of the manuscript that addresses the points raised during the review process.

Please submit your revised manuscript by Mar 17 2024 11:59PM. If you will need more time than this to complete your revisions, please reply to this message or contact the journal office at plosone@plos.org. Please include the following items when submitting your revised manuscript:A rebuttal letter that responds to each point raised by the academic editor and reviewer(s). You should upload this letter as a separate file labeled 'Response to Reviewers'.A marked-up copy of your manuscript that highlights changes made to the original version. You should upload this as a separate file labeled 'Revised Manuscript with Track Changes'.An unmarked version of your revised paper without tracked changes. You should upload this as a separate file labeled 'Manuscript'.

We look forward to receiving your revised manuscript.

Kind regards,

Ayesha Maqbool, PhD

Academic Editor

PLOS ONE

Journal Requirements:

"This research was funded by the central government guides local funds for science and technology development [No. 236Z7201G&No.226Z0302G]; the Special Project of Langfang Key Research and Development under Grant [No. 2023011005B]."

"Acknowledgments

This research was funded by the central government guides local funds for science and tech-nology development No. 236Z7201G ; the Special Project of Langfang Key Research and Devel-opment under Grant No. 2023011005B."

Please be informed that funding information should not appear in the Acknowledgments section or other areas of your manuscript. We will only publish funding information present in the Funding Statement section of the online submission form. 

"This research was funded by the central government guides local funds for science and technology development [No. 236Z7201G&No.226Z0302G]; the Special Project of Langfang Key Research and Development under Grant [No. 2023011005B]."

7. We note that Figure 1 in your submission contain map images which may be copyrighted. All PLOS content is published under the Creative Commons Attribution License (CC BY 4.0), which means that the manuscript, images, and Supporting Information files will be freely available online, and any third party is permitted to access, download, copy, distribute, and use these materials in any way, even commercially, with proper attribution. For these reasons, we cannot publish previously copyrighted maps or satellite images created using proprietary data, such as Google software (Google Maps, Street View, and Earth). For more information, see our copyright guidelines: http://journals.plos.org/plosone/s/licenses-and-copyright.

(1) You may seek permission from the original copyright holder of Figure 1 to publish the content specifically under the CC BY 4.0 license.  

8. We note that Figure 2 in your submission contain copyrighted images. All PLOS content is published under the Creative Commons Attribution License (CC BY 4.0), which means that the manuscript, images, and Supporting Information files will be freely available online, and any third party is permitted to access, download, copy, distribute, and use these materials in any way, even commercially, with proper attribution. For more information, see our copyright guidelines: http://journals.plos.org/plosone/s/licenses-and-copyright.

(1) You may seek permission from the original copyright holder of Figure 2 to publish the content specifically under the CC BY 4.0 license. 

(2) If you are unable to obtain permission from the original copyright holder to publish these figures under the CC BY 4.0 license or if the copyright holder’s requirements are incompatible with the CC BY 4.0 license, please either i) remove the figure or ii) supply a replacement figure that complies with the CC BY 4.0 license. Please check copyright information on all replacement figures and update the figure caption with source information. 

If applicable, please specify in the figure caption text when a figure is similar but not identical to the original image and is therefore for illustrative purposes only.

**Additional Editor Comments:**

This paper introduces a UAV-based approach for multi-target tracking of cattle, demonstrating substantial improvements over traditional methods. The content is well-structured, featuring clear ideas and concise diagrams. However, the author is advised to enhance the paper's quality by providing a more detailed overview of existing research, outlining the strengths and weaknesses of other algorithms. Additionally, in the model design section, a more in-depth explanation of the rationale behind selecting optimization strategies and their application to other models is recommended. 

Reviewers' comments:

Reviewer's Responses to Questions

**Comments to the Author**

1. Is the manuscript technically sound, and do the data support the conclusions?

Reviewer #1: Yes

Reviewer #2: Yes

Reviewer #3: Yes

Reviewer #4: Partly

2. Has the statistical analysis been performed appropriately and rigorously? 

Reviewer #1: N/A

Reviewer #2: Yes

Reviewer #3: N/A

Reviewer #4: Yes

3. Have the authors made all data underlying the findings in their manuscript fully available?

Reviewer #1: No

Reviewer #2: Yes

Reviewer #3: Yes

Reviewer #4: Yes

4. Is the manuscript presented in an intelligible fashion and written in standard English?

Reviewer #1: Yes

Reviewer #2: Yes

Reviewer #3: No

Reviewer #4: Yes

5. Review Comments to the Author

Reviewer #1: The paper is well written. The conclusion section needs to be improved. It should give more insight inot tthe implications of the results and should not duel on summarising the work done.

There is little or no discussion on future work and future directions of the work done. What are the implications? Are there any insights the authors wish to share?

Reviewer #2: This paper presents a method for using unmanned aerial vehicles (UAVs) to achieve multi-target tracking of cattle. Compared with the traditional tracking methods, the performance of the model has been significantly improved in all aspects.

This article has clear ideas, compact structure and beautiful and concise diagrams. However, the author must consider the following suggestions to further improve the quality of his work.

1.In the description of the current state of research, the existing research status should be described in more detail so that the reader can understand the advantages and disadvantages of other algorithms.

2.In the part of model design, I think the author should elaborate in more detail on the reasons for choosing various optimization strategies and whether this strategy has been applied on other models.

3.The author can add some pictures in the necessary places in the article to make the expression more intuitive and easy to understand.

Reviewer #3: The paper falls into the class of Multi-Object-Tracking (MOT) systems for monitoring Holstein cattle. Here are my comments:

1. The English should be proofread. I suggest changing the title of the paper to "High-precision tracking and positioning for monitoring Holstein cattle".

2. The experimental section should be expanded by providing (a) processing time of the approach in terms of FPS; (b) results on MOT71 and MOT20 benchmarks to get a full idea of the general performance of the proposed approach vs. competitors.

Reviewer #4: Precision livestock farming utilises information technology to continuously monitor and manage livestock in real-time, which can improve individual animal health, welfare, productivity and the environmental impact of animal husbandry, contributing to the economic, social and environmental sustainability of livestock farming. Based on artificial intelligence, it promises to driving an agricultural revolution at a time when the world must produce more food using fewer resources.

Thus, Enhanced animal welfare emerges as a pivotal element in contemporary precision animal husbandry, with bovine monitoring constituting a significant facet of precision agriculture. Also, smart drones, outfitted with monitoring systems, have evolved into a viable solution for wildlife protection, moni-toring, and animal husbandry.

It is in this context that in this manuscript the authors propose a tracking algorithm, grounded in deep learning, adhering to the JDT paradigm estab-lished on the CenterTrack algorithm, designed to satisfy the requirements of mul-ti-objective tracking amidst intricate scenarios in practical applications.

The authors believe that the proposed algorithm achieved higher preformances compared to others known in the literature.

The manuscript needs much improvement in order to be accepted for publication. Authors should consider the comments/requirements/questions below:

1). Remarks and suggestions:

* The introduction is nevertheless long, it contains a few sentences that can be skipped.

* Figures 1, a and b) are not clear, they need to be improved in quality.

* The result of the simulation as well as the performance measurements presented in the manuscript must be clearly commented and analyzed. The authors presented surface and generic discussions.

2). Questions

Q1 : The authors state the following sentences (page 13):

"The proposed two-stage matching algorithm consists of two matching stages, both of which ............... thereby preserving the continuity of the tracking trajectory. This process is depicted in Figure 7. "

How did you take this into your program?

Q2 : How can you reduce the likelihood of tracking failures and increase robustness in various lighting conditions?

Q3 : How will you improve the accuracy in the MOT algorithm?

6. PLOS authors have the option to publish the peer review history of their article (what does this mean?). If published, this will include your full peer review and any attached files.

Reviewer #1: No

Reviewer #2: **Yes: **hong tao

Reviewer #3: No

Reviewer #4: No

---

## [Author Response · Author response to Decision Letter 0]

6 Mar 2024

Thank you so much for taking care of our study, and we have carefully revised the manuscript with comprehensive comments from all reviewers. We have incorporated these changes in the revision with YELLOW fonts representing revisions. We like to thank them for their significant contributions to our manuscript. 

 Thank you again for everything you have contributed, and I look forward to your approval.

---

## [Decision Letter · Decision Letter 1]

1 Apr 2024

High-precision tracking and positioning for monitoring Holstein cattle

PONE-D-23-43870R1

Dear Dr. Lue,

We’re pleased to inform you that your manuscript has been judged scientifically suitable for publication and will be formally accepted for publication once it meets all outstanding technical requirements.

Kind regards,

Ayesha Maqbool, PhD

Academic Editor

PLOS ONE

Additional Editor Comments (optional):

Reviewers' comments:

Reviewer's Responses to Questions

**Comments to the Author**

1. If the authors have adequately addressed your comments raised in a previous round of review and you feel that this manuscript is now acceptable for publication, you may indicate that here to bypass the “Comments to the Author” section, enter your conflict of interest statement in the “Confidential to Editor” section, and submit your "Accept" recommendation.

Reviewer #1: All comments have been addressed

Reviewer #2: All comments have been addressed

2. Is the manuscript technically sound, and do the data support the conclusions?

Reviewer #1: Yes

Reviewer #2: Yes

3. Has the statistical analysis been performed appropriately and rigorously? 

Reviewer #1: Yes

Reviewer #2: Yes

4. Have the authors made all data underlying the findings in their manuscript fully available?

Reviewer #1: Yes

Reviewer #2: Yes

5. Is the manuscript presented in an intelligible fashion and written in standard English?

Reviewer #1: Yes

Reviewer #2: Yes

6. Review Comments to the Author

Reviewer #1: (No Response)

Reviewer #2: (No Response)

7. PLOS authors have the option to publish the peer review history of their article (what does this mean?). If published, this will include your full peer review and any attached files.

Reviewer #1: No

Reviewer #2: No

---

## [Editor Report · Acceptance letter]

30 Apr 2024

PONE-D-23-43870R1 

PLOS ONE

Dear Dr. Luo, 

I'm pleased to inform you that your manuscript has been deemed suitable for publication in PLOS ONE. Congratulations! Your manuscript is now being handed over to our production team.

Kind regards, 

on behalf of

Dr. Ayesha Maqbool 

Academic Editor

PLOS ONE